

# Determining the numbers of a landscape architect species (*Tapirus terrestris*), using footprints

Danielle O. Moreira[1,2,3,*], Sky K. Alibhai[1,4,*], Zoe C. Jewell[1,4,*], Cristina J. da Cunha[3], Jardel B. Seibert[2,3] and Andressa Gatti[2,3,*]

[1] Nicholas School of the Environment, Duke University, Durham, NC, USA
[2] Programa de Pós-graduação em Ciências Biológicas (Biologia Animal), Departamento de Ciências Biológicas, Universidade Federal do Espírito Santo, Vitória, Espírito Santo, Brazil
[3] Pró-Tapir, Instituto de Ensino, Pesquisa e Preservação Ambiental Marcos Daniel (IMD), Vitória, Espírito Santo, Brazil
[4] JMP Division, SAS, Cary, NC, USA
* These authors contributed equally to this work.

Corresponding author
Danielle O. Moreira,
daniomoreira@gmail.com

## ABSTRACT

**Background:** As a landscape architect and a major seed disperser, the lowland tapir (*Tapirus terrestris*) is an important indicator of the ecological health of certain habitats. Therefore, reliable data regarding tapir populations are fundamental in understanding ecosystem dynamics, including those associated with the Atlantic Forest in Brazil. Currently, many population monitoring studies use invasive tagging with radio or satellite/Global Positioning System (GPS) collars. These techniques can be costly and unreliable, and the immobilization required carries physiological risks that are undesirable particularly for threatened and elusive species such as the lowland tapir.

**Methods:** We collected data from one of the last regions with a viable population of lowland tapir in the south-eastern Atlantic Forest, Brazil, using a new non-invasive method for identifying species, the footprint identification technique (FIT).

**Results:** We identified the minimum number of tapirs in the study area and, in addition, we observed that they have overlapping ranges. Four hundred and forty footprints from 46 trails collected from six locations in the study area in a landscape known to contain tapir were analyzed, and 29 individuals were identified from these footprints.

**Discussion:** We demonstrate a practical application of FIT for lowland tapir censusing. Our study shows that FIT is an effective method for the identification of individuals of a threatened species, even when they lack visible natural markings on their bodies. FIT offers several benefits over other methods, especially for tapir management. As a non-invasive method, it can be used to census or monitor species, giving rapid feedback to managers of protected areas.

## INTRODUCTION

One of the key ecological processes that maintain the health status of tropical forests is seed dispersal (*Boissier et al., 2014*). Vertebrates can disperse between 45% and 90% of

woody species seeds and mammals are important dispersers in warmer forests (*Almeida-Neto et al., 2008*). The lowland tapir, *Tapirus terrestris* (*Linnaeus, 1758*), a remaining species of the Pleistocene megafauna in South and Central America (*Simpson, 1980*; *Eisenberg, 1981*), is a large selective browser (180–300 kg), and an important vertebrate seed disperser because it can disperse large-seeded species over long distances (*Dirzo & Miranda, 1991*; *Galetti et al., 2001*; *O'Farrill et al., 2012*; *Bueno et al., 2013*).

In Brazil, lowland tapir populations are decreasing at an alarming rate due to habitat destruction and fragmentation, illegal fires (*Chiarello, 2000a*; *Michalski & Peres, 2007*), hunting (*Chiarello, 2000b*; *Peres, 2000*), and road kill (*Medici & Desbiez, 2012*; *Souza, Cunha & Markwith, 2015*), especially in the Atlantic Forest and Cerrado ecosystems. Consequently, the lowland tapir is listed as vulnerable in the IUCN Red List of Threatened Species (*Naveda et al., 2008*), and in the Brazilian list of Threatened Species (*Instituto Chico Mendes de Conservação da Biodiversidade, 2016*). In the Atlantic Forest, tapirs are considered as Endangered (*Medici et al., 2012*).

The disappearance of frugivores such as tapirs from disturbed tropical forests can reduce the recruitment and survival of plant species, threatening forest regeneration (*Boissier et al., 2014*). Thus, tapirs can be considered a landscape architect and a major indicator of ecological health of landscapes (*Bueno et al., 2013*; *O'Farrill, Galetti & Campos-Arceiz, 2013*; *Giombini, Bravo & Tosto, 2016*). Understanding tapir population dynamics is therefore crucial to the health of Brazilian Atlantic Forest ecosystems.

*Gatti, Brito & Mendes (2011)* and *Medici & Desbiez (2012)* estimated that a lowland tapir population is considered genetically stable in the Atlantic Forest if it consists of at least 200 animals at all times over a 100-year period. However, only three areas currently have a population of more than 200 individuals in the Atlantic Forest (*Medici et al., 2012*), the Serra do Mar forest complex, the State Park of Rio Doce and the Linhares/Sooretama forest complex. If no conservation intervention is taken to protect the species in the Atlantic Forest, it is predicted that the remaining populations will disappear in 33 years (*Medici et al., 2012*). Without reliable information on demographics, developing a strategy for sustaining free-ranging tapir populations will be impossible (*Foerster & Vaughan, 2002*). However, while lowland tapir population censusing and monitoring is essential, it is a difficult task because tapir are mostly crepuscular and nocturnal (*Oliveira-Santos et al., 2010*; *Wallace, Ayala & Viscarra, 2012*; *Cruz et al., 2014*).

Classical studies of tapir population censusing and monitoring include individual counting by direct sightings (*Fragoso, 1987*; *Glanz, 1990*), scat surveys (*Naranjo & Cruz, 1998*; *Flesher, 1999*; *Ramírez, 2013*), footprint counts (*Naranjo & Cruz, 1998*; *Flesher, 1999*; *Naranjo & Bodmer, 2007*), line transect sampling (*Naranjo & Cruz, 1998*; *Naranjo & Bodmer, 2007*; *Medici, 2010*; *Wallace, Ayala & Viscarra, 2012*), the use of high technology such as camera traps (*Wallace, Ayala & Viscarra, 2012*; *Carbal-Borges, Godínez-Gómez & Mendonza, 2014*; *Mejía-Correa, Diaz-Martinez & Molina, 2014*; *Tobler et al., 2014*), and Very High Frequency (VHF)/Global Positioning System (GPS) collars (*Foerster & Vaughan, 2002*; *Medici, 2010*). However, the research costs and duration have often hampered the implementation of these methods. Nevertheless, many facets of wildlife

research demand the recognition of individual animals (*Alibhai, Jewell & Law, 2008*). For tapirs, each of these methods has disadvantages. For example, adult tapirs do not have natural coat patterns that can easily distinguish an individual during a census or in a camera trap image and to fit a tag or radio-collar, it is necessary to capture and immobilize a tapir, which comes with ongoing expenses and risks to animal and researcher.

A new cost-effective and non-invasive method for identifying endangered species is the footprint identification technique (FIT) (*Alibhai, Jewell & Law, 2008*; *Jewell & Alibhai, 2013*; *Jewell et al., 2016*; *Alibhai, Jewell & Evans, 2017*), which is capable of identifying at the individual, sex and age-class levels from digital images of footprints of certain species. It is considered a cost-effective method because of the low investment for fieldwork combined with high accuracy in individual identification (*Gusset & Burgener, 2005*; *Jewell, 2013*; *Pimm et al., 2015*; *Jewell et al., 2016*). In this way, footprints can serve as an alternative to natural coat patterns or body marks for those species whose foot is of sufficient complexity to create a footprint with individual characteristics. The technique proved to be effective with footprints of black rhino (*Diceros bicornis*) (*Jewell & Alibhai, 2001*) and white rhino (*Ceratotherium simium*) (*Alibhai, Jewell & Law, 2008*). FIT has subsequently been adapted for several other species, including Amur tiger (*Panthera tigris* ssp. *altaica*) (*Gu et al., 2014*), cougar (*Puma concolor*) (*Jewell, Alibhai & Evans, 2014*; *Alibhai, Jewell & Evans, 2017*), brown bear (*Ursus arctos*) (*Petridou, Sgardelis & Youlatos, 2008*), cheetah (*Acinonyx jubatus*) (*Jewell et al., 2016*), panda (*Ailuropoda melanoleuca*) (*Li et al., 2018*), Bengal tiger (*Panthera tigris* ssp. *tigris*), polar bear (*Ursus maritimus*), and Baird's tapir (*Tapirus bairdii*) (*Jewell & Alibhai, 2013*). In 2007, the method was tested for the first time for the lowland tapir in the Atlantic Forest (*Medici, 2010*).

This study is an attempt to census one of the last regions with a viable population of tapirs in the southeastern Atlantic Forest—the Linhares/Sooretama forest complex in the state of Espírito Santo, Brazil. This population survives in a forested area of about 50,000 ha. FIT methodology has the potential to census tapir populations and help implement species management and conservation strategies, especially for disturbed tropical ecosystems dependent on frugivores for maintenance. Thus, determining tapir distribution and numbers accurately is a major challenge for tropical ecologists.

## MATERIALS AND METHODS

### Study area

We performed the study in the Private Natural Heritage Reserve Recanto das Antas (*Reserva Particular do Patrimônio Natural Recanto das Antas*, in Portuguese, hereafter "RPPNRA") and adjacent private areas to the reserve (Fig. 1). The RPPNRA is a private protected area of 2,212 ha owned by Fibria Celulose S.A., a Brazilian company, located in the Linhares/Sooretama forest complex, north of the state of Espírito Santo, Brazil. It is contiguous with the 27,858 ha Sooretama Biological Reserve and the 22,711 ha Vale Natural Reserve. The RPPNRA is located at the coordinates 19°05′ south latitude, 39°58′ west longitude, and mostly consists of discontinuous primary vegetation, interposed by extensive eucalyptus and papaya plantations, cabruca (cacao trees planted in the shade of thinned native forest), seringal (rubber tree culture), and
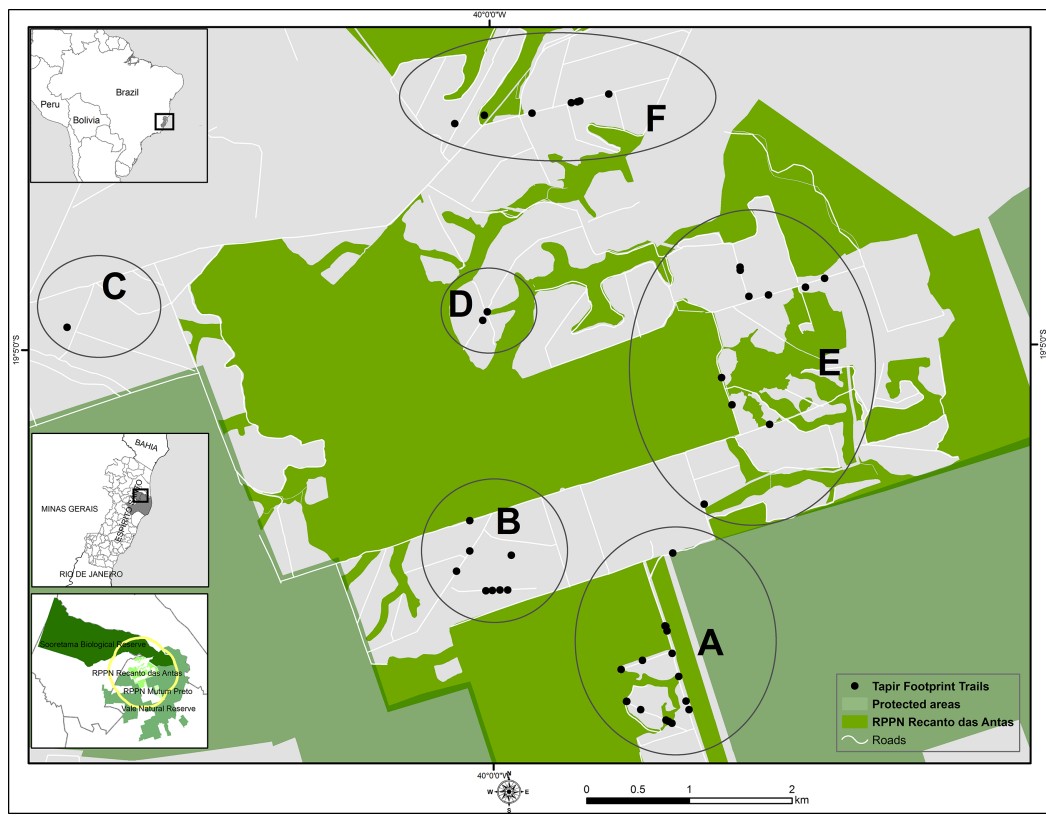

**Figure 1 Survey sites for lowland tapir footprints.** Location of the main survey sites (A–F) used for lowland tapir footprints survey in the Private Natural Heritage Reserve Recanto das Antas and surroundings, Espírito Santo, Brazil. Cartographic base source: protected areas—*Cadastro Nacional de Unidades de Conservação (2015)*; political boundaries—*Instituto Brasileiro de Geografia e Estatística (2001, 2005)*. Datum: World Geodetic System 1984 (WGS 1984). Coordinate system/projection: Universal Transverse Mercator (UTM) 24S.

smaller amounts of coffee plantations and cattle pastures, which are not part of the reserve (*Centoducatte et al., 2011*). Unpaved roads used for eucalyptus harvesting and transportation also cross the area.

The study site is situated in an area of Barreiras Formation (Tertiary sediments, or "Tabuleiros"), where the vegetation is part of the Ombrophilous Forest Region of the Lowlands (*Instituto Brasileiro de Geografia e Estatística, 1987*), also known as the Tabuleiros Atlantic Forest (*Rizzini, 1997*). The Tabuleiros Forest of Linhares and Sooretama region is considered of high biological importance for the conservation of the biodiversity (*Conservação Internacional do Brasil et al., 2000*), a priority area for the conservation of medium- and large-sized mammals (*Galetti et al., 2009*) and part of the UNESCO World Heritage Discovery Coast Atlantic Forest Reserves.

## Collection of footprints

The lowland tapir, an ungulate from the order Perissodactyla, whose toes are surrounded by a hoof, has four digits on the forefeet, representing anatomical digits 2, 3, 4, and 5. The smallest one (the fifth digit) appears only in footprints impressed in soft ground

(*Ballenger & Myers, 2001*). The hind feet have only three digits (anatomical digits 2, 3, and 4), and they each appear in the footprint (*Medici, 2011*). Tapirs, like many terrestrial large mammals, typically register the hind foot in the impression left by the front foot when they are walking (*Jewell & Alibhai, 2001*; *Elbroch, 2003*; *Alibhai, Jewell & Law, 2008*; *Jewell et al., 2016*).

Footprint surveys were performed on 17 dirt (unpaved, soft surface) roads, which represent the main dirt roads crossing the study area. The total length of surveyed roads was 35,118 m. One road was visited at least once in each field survey. We chose dirt roads because (1) tapirs use them frequently, (2) the roads cross many different types of habitat, including forest and agriculture land, and (3) we had easy access to them. Because lowland tapirs are active mostly between dusk and dawn (*Oliveira-Santos et al., 2010*; *Wallace, Ayala & Viscarra, 2012*; *Cruz et al., 2014*), our surveys were done early in the morning and in the late afternoon, when there was sufficient light for photography. The availability of footprints of tapirs on roads also depended to some extent on the weather. The quality of footprints was sometimes reduced when the weather was too dry because footprint impressions were not held so well by the substrate, and when it rained footprints were lost. The FIT requires well-defined clear footprints, generally obtained from animals walking at a relaxed pace so we used only undistorted footprints in this study.

We collected sets of digital images of footprints from wild lowland tapirs, as follows, following the WildTrack FIT protocol (*Alibhai, Jewell & Law, 2008*; *Jewell et al., 2016*). First, we identified a trail (an unbroken series of footprints made by the same animal) of footprints left by an individual (Fig. 2A). Left hind footprints were used for the analysis following the protocol given in *Alibhai, Jewell & Law (2008)*. For each footprint image, a scale (in centimeters) was placed to the left and bottom of it, and a slip was placed alongside one of the rulers with the date, name of the road, UTM coordinates, collector, and footprint code (Fig. 2B). Each day, every trail and footprint received a unique code. Photographs were taken at high resolution (2,248 × 4,000 pixels or higher), from directly overhead (although FIT can work quite effectively at 1,600 × 1,200).

To avoid the risk of collecting a footprint more than once, we obliterated each footprint after taking the photos. We carried out 10 fieldwork sessions of three to five days duration during 10 months between March 2014 and June 2015 in the study area. To enable a more detailed analysis of the minimum number of tapirs in the area, we arbitrarily divided the study area into six locations based on distribution of footprint trails (A–F; Fig. 1). The subdivision of locations allowed a comparison of the number of trails and the number of animals in different areas. The map was designed in ArcMap 10.1 (Esri, Redlands, CA, USA). The farthest distance between two areas was 6,297 m (area C and E), the smallest distance was 1,110 m (area A and B), while the average distance among the areas was 3,290 m (Table 1).

We received permission to conduct fieldwork in our study area from SISBIO/Instituto Chico Mendes de Conservação da Biodiversidade (number 32565-5), according to Brazilian laws.

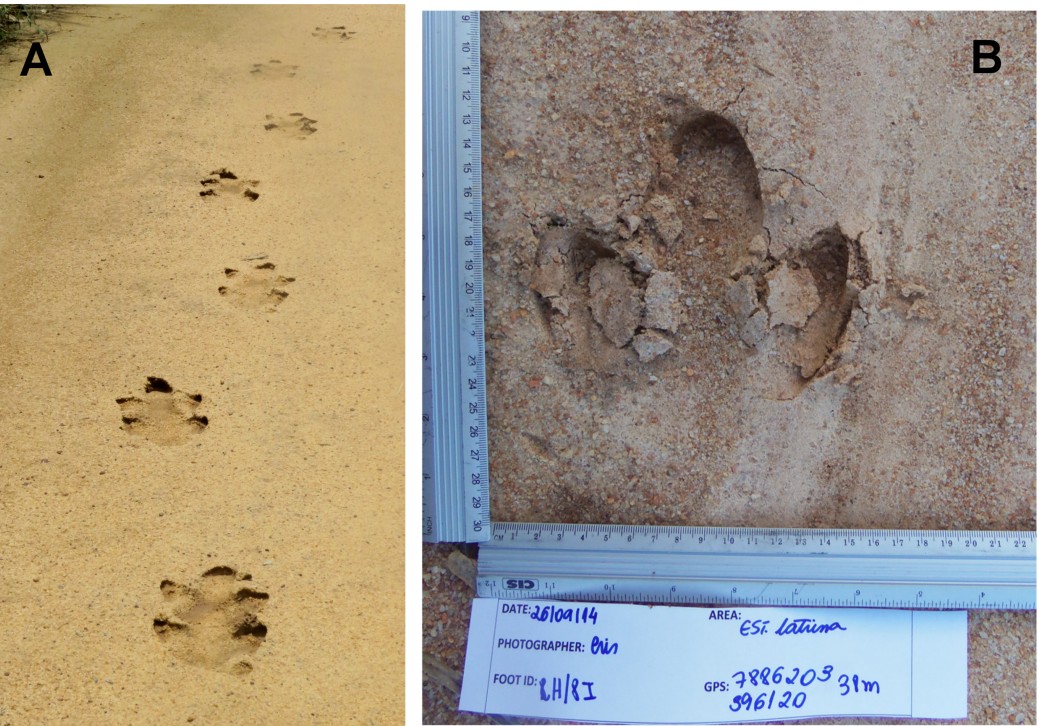

**Figure 2 Lowland tapir trail and footprint.** (A) Trail of lowland tapir (*Tapirus terrestris*) along a dirt road. Photo credit: Danielle O. Moreira. (B) Lowland tapir left hind footprint photo collected during footprint surveys. Photo credit: Cristina J. da Cunha.

**Table 1 Approximate distances (in meters) between sampling sites.**

|   | A | B | C | D | E | F |
|---|---|---|---|---|---|---|
| **A** | 0 | | | | | |
| **B** | **1110** | 0 | | | | |
| **C** | 6130 | 4245 | 0 | | | |
| **D** | 2893 | 1837 | 3872 | 0 | | |
| **E** | 1567 | 2547 | **6297** | 2370 | 0 | |
| **F** | 4494 | 3868 | 4203 | 1860 | 2058 | 0 |

Notes:
Study area is located in the Private Natural Heritage Reserve Recanto das Antas and surroundings, Espírito Santo, Brazil. In bold are the smallest and farthest distances between two sites. See map in Fig. 1 for sites reference.

## Analysis

The identification of individuals using FIT is based on the morphometrics of the footprint (*Alibhai, Jewell & Towindo, 2001*; *Alibhai, Jewell & Law, 2008*; *Jewell et al., 2016*; *Alibhai, Jewell & Evans, 2017*). Because each species has a unique foot anatomy, FIT algorithms are designed to be species specific. Each species FIT algorithm defines the footprint measurements that allow the software to discriminate between individuals for that species (*Alibhai, Jewell & Evans, 2017*). The lowland tapir algorithm was developed by S. K. Alibhai and Z. C. Jewell (Document S1, Document S2) and *Medici (2010)* through the collection of a training-set database of footprint photographs from
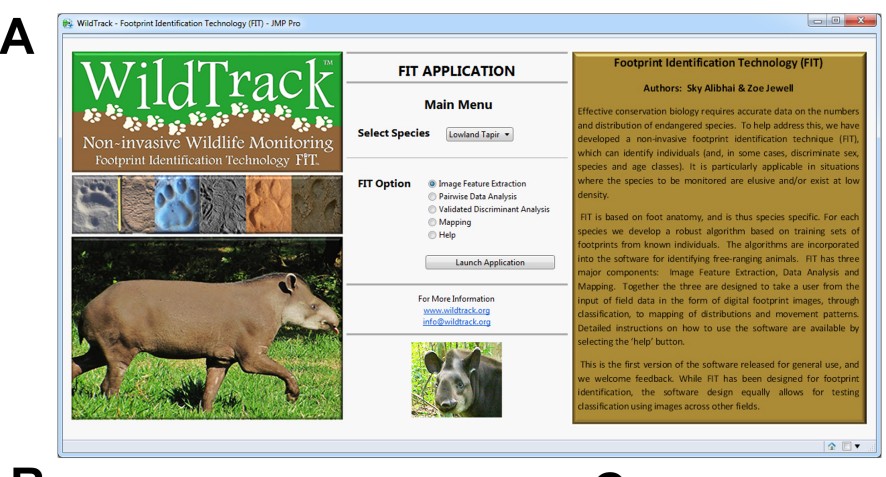

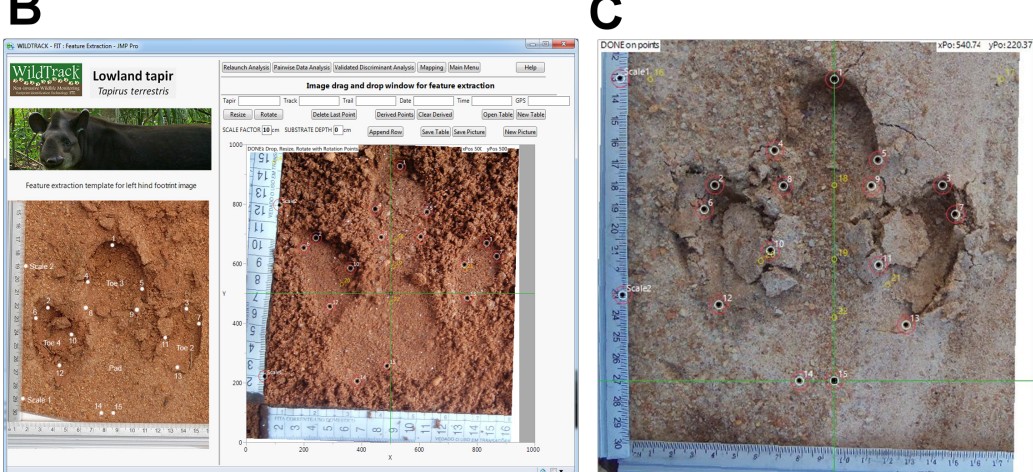

**Figure 3** **Footprint identification technique (FIT).** (A) Main menu window for lowland tapir in FIT. (B) Feature extraction window in FIT for lowland tapir. (C) Footprint image after processing using FIT script. Landmark points are in black and derived points in yellow. Image and photos credits: Danielle Moreira/Andressa Gatti/WildTrack.

known individual captive tapir (Table S1). Randomized holdback trials using the FIT algorithm with the training set gave accuracies of 90% for the predicted number of individuals (for more details about the FIT algorithm, see *Jewell et al. (2016)* and *Alibhai, Jewell & Evans (2017)*). The FIT software home-page, feature extraction page and footprint variable extraction are shown in Figs. 3A and 3B. The training-set library of 426 images from 36 captive individuals was used to extract the algorithm for individual identification of tapirs using FIT (*Alibhai, Jewell & Towindo, 2001*; *Alibhai, Jewell & Law, 2008*; *Medici, 2010*; *Jewell et al., 2016*; *Alibhai, Jewell & Evans, 2017*). To census the unknown tapir population in this study, we then applied this previously-derived algorithm to the analysis of new footprint images collected from our study area.

Prior to the statistical analysis, we uploaded each footprint image into GIMP 2.8.14 software (*GIMP team, 2014*) to optimize color contrast and crop the images. Each image was then rotated to a standardized orientation using the FIT add-in in JMP Pro 12

(SAS, Cary, NC, USA). Fifteen anatomically-based landmark points on a tapir footprint were prior chosen that could be repeated in other studies and clearly identified (*Alibhai, Jewell & Law, 2008*; *Jewell et al., 2016*). Using FIT software, these landmark points were then manually placed on each footprint image using cross-hair guidelines to minimize bias (Figs. 3B and 3C). From these 15 landmark points, FIT script defined a further set of seven derived points, geometrically constructed from the set of landmarks points (*Jewell et al., 2016*). From the 15 landmark points and seven derived points, a total of 121 measurements (distances, angles, and areas) were generated for each tapir footprint.

This full set of measurements (the 'geometric profile'; Table S2) was taken to include all those that might prove useful in discriminating between footprints (Figs. 3B and 3C). The set of measurements of all the tapir footprints constituted the dataset upon which all FIT analyses were performed (*Alibhai, Jewell & Law, 2008*; *Jewell et al., 2016*). A full step-by-step video account of the FIT is reported in *Jewell et al. (2016)*.

Footprint identification technique is based on a comparison of sets of footprints (trails) where each trail is an unbroken set of footprints made by one individual. The comparison is made using a customized robust cross-validated pair-wise discriminant analysis model. Where trails were composed of more than 10 footprints, they were randomly divided into sub-trails each consisting of 5–8 footprints. For example, trail 8214 was divided into three sub-trails arbitrarily—A8214A, A8214B & A8214C—with the prefix (A) denoting the location of the trail in the study area (see Fig. S1), and the suffixes (A), (B), and (C) represent the three sub-trails (see *Jewell & Alibhai, 2001*; *Alibhai, Jewell & Law, 2008*; *Jewell et al., 2016*). The sub-trails from a single unbroken trail were named 'self' sub-trails for purposes of classification. Sub-trails from different trail sets were named 'non-self' sub-trails. This enabled the comparison of self sub-trails and non-self sub-trails during FIT analysis.

For the whole study area (7,900 ha), we analyzed the data at several different scales: (A) We used all the sub-trails to identify the number of individuals in the entire study area (pooled data), (B) we analyzed the data for each location (A–F) separately and we summed that data to identify the number of individuals, and (C) we paired the locations and once again compared the pooled data with summed data. In doing so, we were able to identify if one or more tapirs were visiting more than one location. Finally, using the data from the six different locations (A–F), we tested the relationship between the numbers of trails and tapir population estimates. Images were processed in the JMP data visualization software.

## RESULTS

We collected a total of 547 footprint images from 48 trails with an average of 11.40 footprints per trail, but after discarding some poor quality images we used 440 footprints from 46 trails in the analysis (an average of 9.57 footprints per trail). The minimum number of usable footprints in a trail was four and maximum was 23. The data were subjected to FIT analysis which generates a cluster dendrogram giving a prediction for the estimated minimum number of individuals and the relationship between sub-trails

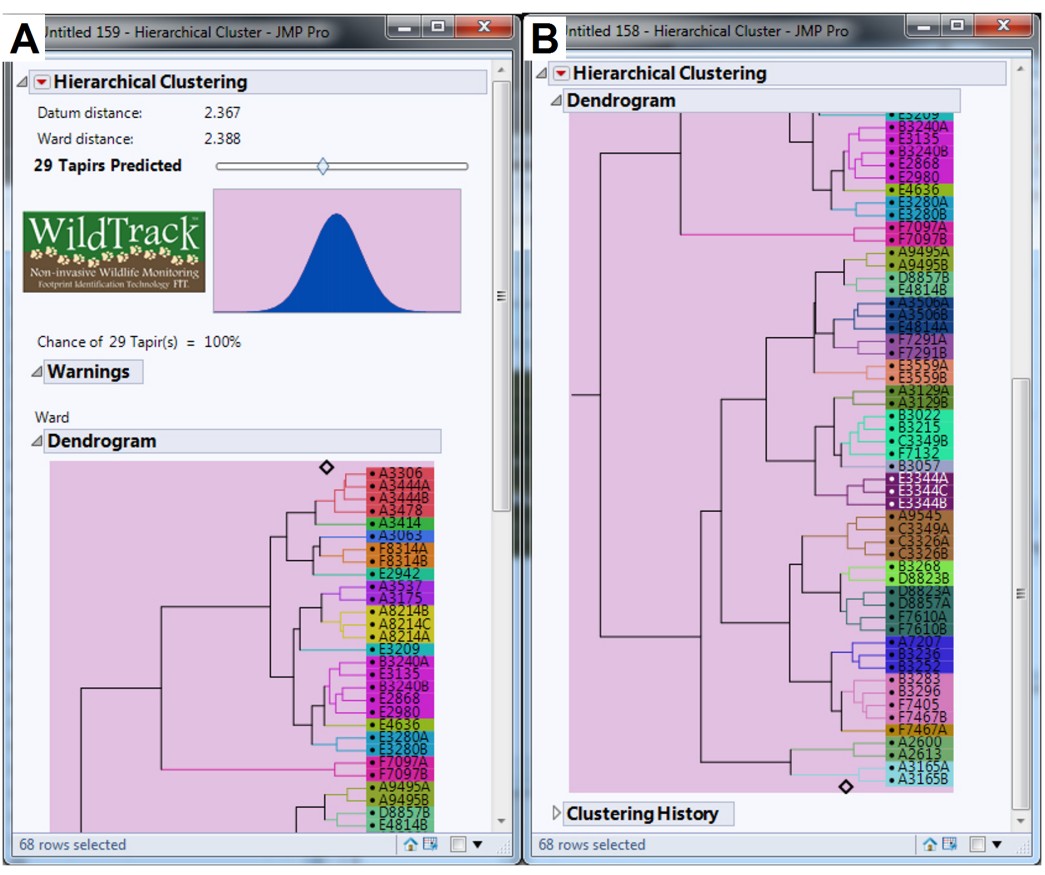

**Figure 4 Dendrogram generated by FIT algorithm predicting the minimum number of lowland tapir (*Tapirus terrestris*) for the studied area.** (A) Shows 14 tapirs identified. (B) Shows more 15 tapirs identified. Surveys ranged from location (A) to location (F) (see Fig. 1 for site references) in the Private Natural Heritage Reserve Recanto das Antas and surroundings, in Espírito Santo, Brazil. Each letter represents a location; a number represents a trail; a number and letter, a sub-trail; and the colors represent one individual.                                

(*Jewell et al., 2016*). For the whole study area, FIT gave an estimate of 29 different individuals for pooled data (Figs. 4A and 4B). We then analyzed the data for each of the six locations and the summed estimate for all six locations was 35 individuals. Finally, we compared the pooled and summed estimates for different combinations of locations (Table 2). All pooled estimate values were either equal to or lower than summed values. Location (A) had the most tapirs identified (*n* = 12; see Fig. S1), whereas locations (C) and (D) (see Fig. S2) had only one individual identified (Table 2). Locations (A) and (E) combined represented more than 65% of total individuals identified in the study area (Table 2; Figs. S1 and S3). FIT identified six and seven individuals for the locations (B) and (F), respectively (see Figs. S4 and S5).

The difference between the pooled (29) and summed (35) estimates indicates that six individuals appeared in more than one area during the study period (Table 2). For example, three individual used the closest sites A + B (1,110 m apart). One individual visited the sites A + E (1,567 m apart), three tapirs were recorded visiting the sites B + F

**Table 2 Number of lowland tapirs identified in one or more sites using the footprint identification technique (FIT).**

| Location | No. of footprints | No. of trails | No. of sub-trails | No. of footprints per sub-trail | Estimated No. of tapirs |
|---|---|---|---|---|---|
| A | 150 | 16 | 23 | 5–8 | 12 |
| B | 71 | 9 | 10 | 6–8 | 6 |
| C | 20 | 2 | 4 | 5 | 1 |
| D | 20 | 2 | 4 | 5 | 1 |
| E | 99 | 10 | 15 | 5–8 | 8 |
| F | 80 | 7 | 12 | 6–8 | 7 |
| **Total for study site** | **440** | **46** | **68** | **5–8** | **35 (29)** |
| A + B | 221 | 25 | 33 | 5–8 | 18 (15) |
| A + C | 170 | 18 | 27 | 5–8 | 13 (13) |
| A + D | 170 | 18 | 27 | 5–8 | 13 (13) |
| A + E | 249 | 26 | 38 | 5–8 | 20 (19) |
| A + F | 230 | 23 | 35 | 5–8 | 19 (18) |
| B + C | 91 | 11 | 14 | 5–8 | 7 (7) |
| B + D | 91 | 11 | 14 | 5–8 | 7 (7) |
| B + E | 170 | 19 | 25 | 5–8 | 13 (13) |
| B + F | 151 | 16 | 22 | 6–8 | 13 (10) |
| C + D | 40 | 4 | 8 | 5 | 2 (2) |
| C + E | 119 | 12 | 19 | 5–8 | 9 (9) |
| C + F | 100 | 9 | 16 | 5–8 | 8 (8) |
| D + E | 119 | 12 | 19 | 5–8 | 9 (9) |
| D + F | 100 | 9 | 16 | 5–8 | 8 (8) |
| E + F | 179 | 17 | 27 | 5–8 | 15 (15) |

**Notes:**
Study area is located in the Private Natural Heritage Reserve Recanto das Antas and surroundings, Espírito Santo, Brazil. See map in Fig. 1 for sites reference. Estimates of numbers of lowland tapirs are for summed data. Pooled data are in parenthesis.

(3,868 m apart) and two individuals used the sites A + F (4,494 m apart). The other sites, when combined, did not indicate any individuals visiting more than one area.

Finally, although the data were limited to just six different locations (A–F), we examined the relationship between the number of trails (predictor variable $x$) and the tapir population estimate (response variable $y$) for each location (Fig. 5). The regression was highly significant ($y = 1.24 + 0.62x$, $R^2 = 0.9660$, $p < 0.001$).

## DISCUSSION

We have demonstrated a practical application of FIT as a means of monitoring the health of the Atlantic Forest of Brazil, through an assessment of the numbers of an indicator species, the lowland tapir. In the 10 months over which this census was conducted, we identified at least 29 different individuals of lowland tapir in the RPPNRA and surroundings. The study suggested that tapirs in the RPPNRA might have overlapping ranges and that FIT could identify that the some individuals visited more than one location.

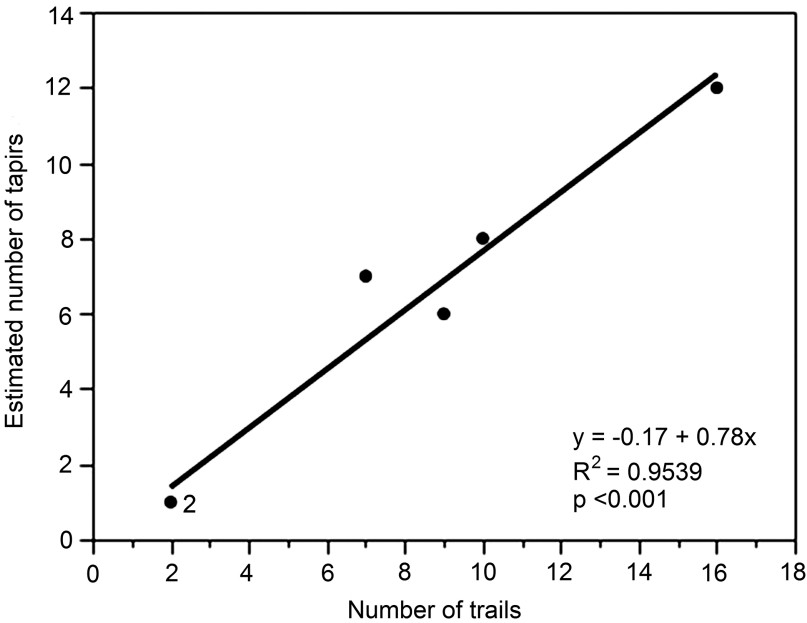

**Figure 5 Relationship between numbers of trails and estimated number of unique individuals of lowland tapirs as determined by FIT in the study site.** The figures for numbers of trails are derived from the different locations (A–F).

Lowland tapirs exhibit extensive home range overlap (*Noss et al., 2003*; *Medici, 2010*). In our study, we estimated that at least six different individuals shared the same areas. In addition, we found evidence to suggest that they moved long distances within the study area. Lowland tapirs can easily traverse low-quality and non-natural habitats, moving through the landscape matrix in between patches of forest, including eucalyptus and agriculture fields (*Noss et al., 2003*; *Medici, 2010*; *Centoducatte et al., 2011*). In our study area, a landscape composed basically of agriculture, eucalyptus forest, pasture, and secondary patches of Tabuleiro Forest, we identified footprints of the same individuals in at least two sites, from 1,110 to 4,494 m apart. This result indicates that the landscape matrix in the RPPNRA area provides a certain level of functional connectivity that allow tapirs to mediate, considerably, the gene flow of many plant species since they have a key role in the dispersion of many seeds (*Tobler, Carrillo-Percastegui & Powell, 2009*; *Bueno et al., 2013*; *O'Farrill, Galetti & Campos-Arceiz, 2013*; *Giombini, Bravo & Tosto, 2016*). This ability of tapir to travel among heterogeneous habitats, dispersing seeds over distances is essential for plants that depend on long dispersion to survive (*Giombini, Bravo & Tosto, 2016*).

The footprint identification technique offers several substantial benefits over other monitoring methods, especially those that are invasive to the study species (*Alibhai, Jewell & Evans, 2017*). First, the footprints of tapirs are easy to find and very abundant. With appropriate weather conditions and substrates, and a moderate sampling effort, footprints can be used effectively for censusing tapir populations. The RPPNRA and its surroundings have more than 70 dirt roads, and all the roads used for this study (a total of 17) were visited by lowland tapirs at least once. Most of these unpaved roads have a good substrate for

footprints and trails (e.g., sand or clay substrate). They are located in different parts of the studied area, covering different habitats, and footprint trails found were mostly long and well defined. Also, it is likely that tapirs use dirt roads more frequently than off roads (*Di Bitetti, Paviolo & De Angelo, 2014*). In our study area, footprints of tapir are more visible and frequent in those unpaved roads than in other areas inside the forest, such as game trails. To find a footprint or a long trail of footprints in the forest is a difficult task, especially because of the great amount of litter that cover the forest floor.

Second, an invasive method (i.e., immobilization and capture) carries a small risk of individual mortality, and it is possible that immobilization itself may negatively impact on female fertility (*Alibhai, Jewell & Towindo, 2001*). In addition, instrumented animals may exhibit changed behavior that is not representative of the population as a whole, and therefore poses questions about resulting data reliability (*Jewell & Alibhai, 2013*; *Jewell, 2013*). Third, a systematic FIT survey can be carried out in a relatively short period of time depending on personnel and resources. Fourth, FIT projects can employ local trackers' expertise to locate tapirs and other species' footprints, supporting the local people and their valuable traditional knowledge. Lastly, FIT can be used alongside camera-trapping or line transects as a powerful addition to the monitoring toolbox. An additional finding was the significant relationship between the numbers of trails and the estimated uniquely identified individuals of tapir that suggests that this could be used as an effective method for estimating tapir numbers by simply counting the number of trails. However, we believe that this needs to be validated in different study sites to include detection probabilities before it is employed as a useful index for tapir censusing as it is likely to be influenced by sampling procedure.

## CONCLUSION

This is the first census survey attempt using footprints for the Tabuleiros Forest, and we will continue to apply the FIT method to census populations and individuals over time in the RPPNRA and surrounding areas. We hope that this will form the basis of a long-term study that we can replicate in nearby sites to estimate the species population, and determine its status and viability in the Linhares/Sooretama forest complex region.

The FIT software can be made available free of charge (http://www.wildtrack.org), and sits as an add-in to JMP statistical analysis software which is available commercially (https://www.jmp.com). We help non-profits and other groups with demonstrable need to apply for a friends of JMP free annual license. We offer in-situ FIT training workshops for users on request and usually hosted in conjunction with a local partner.

Research with this type of methodology is needed to improve our ability to manage and conserve tapirs. This technique is able to rapidly provide data on the numbers and distribution of a key seed-disperser species, from footprints alone. A non-invasive and low-cost method, like FIT, is essential to collect data on populations of threatened species, and provide those data to managers of protected areas. Our surveys within a heterogeneous landscape, such as the RPPNRA, with the identification of a minimum of 29 tapir individuals, confirms the conservation value of this area as a stronghold for populations of *T. terrestris*.

## ACKNOWLEDGEMENTS

The authors would like to acknowledge the collaboration of Dr. Patrícia Medici, Dr. Stuart Pimm, Mr. Diego Edon, B.Sc. Paula Modenesi Ferreira and B.Sc. Amabili Falqueto Mistura. The authors also would like to thank Fazenda Cupido & Refúgio and the JMP division (https://www.jmp.com) of SAS (https://www.sas.com) USA for their generous support.

### Funding

This work was supported by Fibria Celulose S.A. and Idea Wild. Danielle O. Moreira received a fellowship from Conselho Nacional de Desenvolvimento Científico e Tecnológico/Ciências sem Fronteiras (CNPq/CsF process number 249600/2013-7). There was no additional external funding received for this study. The funders had no role in study design, data collection and analysis, decision to publish, or preparation of the manuscript.

### Grant Disclosures

The following grant information was disclosed by the authors:
Fibria Celulose S.A.
Idea Wild.
Conselho Nacional de Desenvolvimento Científico e Tecnológico/Ciências sem Fronteiras (CNPq/CsF): 249600/2013-7.

### Competing Interests

The authors declare that they have no competing interests.

### Author Contributions

- Danielle O. Moreira conceived and designed the experiments, performed the experiments, analyzed the data, contributed reagents/materials/analysis tools, prepared figures and/or tables, authored or reviewed drafts of the paper, approved the final draft.
- Sky K. Alibhai conceived and designed the experiments, performed the experiments, analyzed the data, contributed reagents/materials/analysis tools, prepared figures and/or tables, authored or reviewed drafts of the paper, approved the final draft.
- Zoe C. Jewell conceived and designed the experiments, performed the experiments, analyzed the data, contributed reagents/materials/analysis tools, authored or reviewed drafts of the paper, approved the final draft.
- Cristina J. da Cunha performed the experiments, contributed reagents/materials/analysis tools, authored or reviewed drafts of the paper, approved the final draft.
- Jardel B. Seibert performed the experiments, contributed reagents/materials/analysis tools, authored or reviewed drafts of the paper, approved the final draft.
- Andressa Gatti conceived and designed the experiments, performed the experiments, contributed reagents/materials/analysis tools, authored or reviewed drafts of the paper, approved the final draft.

## Field Study Permissions

The following information was supplied relating to field study approvals (i.e., approving body and any reference numbers):

We received permission to conduct fieldwork from SISBIO/Instituto Chico Mendes de Conservação da Biodiversidade (number 32565-5).

## Data Availability

The raw data are provided in Supplemental Dataset Files.

## Supplemental Information

Supplemental information for this article can be found online at http://dx.doi.org/10.7717/peerj.4591#supplemental-information.

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
