# Peer review of "Determining the numbers of a landscape architect species (Tapirus terrestris), using footprints"

_PeerJ, doi:10.7717/peerj.4591_

## Round 0.1 · original submission · Minor Revisions

Both reviewers recommend reconsideration of your paper following minor revision, and I agree with them. I invite you to resubmit your manuscript after addressing ALL reviewer comments.

When resubmitting your manuscript, please carefully consider all issues mentioned in the reviewers' comments, outline every change made point by point, and provide proper rebuttals for any comments not addressed.

Although the language is absolutely fine in most of the paper, the referee 2 found several small mistakes and typos. Therefore, I believe the updated manuscript needs to be reviewed by a native English speaker before resubmission.

Reviewer 1 ·

Basic reporting

The article generally meets these requirements. I suggest some minor revisions below, which I would expect the authors would easily implement, and which address any concerns on reporting.

Line 42: ‘the South and central America’; delete ‘the’
Line 99: “for censusing’ or “to census’
Line 150: ‘under’ the footprint is not what you literally mean, as Fig. 2 shows. Perhaps, ‘alongside one of the ruler’ is clearer.
Line 170: about ‘the’ FIT algorithm
Line 180: what level of accuracy was achieved with the test data? This fact is an important detail.
Lines 183 – 195: This text is a little confusing. As I understand it, 15 landmarks are chosen on anatomical grounds to yield repeatable measurements between these landmarks. A script then yields a further seven derived points and a total of 121 measurements based on these 22 points are computed. These 121 measurements are intended to include all measurements that might prove useful in discrimination. All of this procedure occurs before any data, other than a generic tapir footprint, is examined. The training data is then employed to select the measurements amongst the 121 that are most effective at discrimination, yielding the ‘tapir’ FIT algorithm, which can then be applied to field data. The construction of the algorithm needs to be more clearly separated from its application to field data.
Line 206: ‘at’ several different scales
Line 223: ‘more’ than what? Or do you mean area A had the most tapirs?
Line 225: ‘represented’ rather than ‘presented’
Line 226: State the results for areas B and F here, as well as referring the reader to the Supplementary Figures.
Lines 231 – 232: ‘The other sites, when combined, did not indicate any individuals visiting more than one area.’

Line 235: Explicitly indicate what the response variable y is (presumably number of tracks) and the predictor variable x is (presumably number of tapirs). Replace ‘Rsq’ by ‘R2’

Lines 240 -242: This statement is too strong. Rather, you are relying on FIT’s identifications too infer tapirs visit several locations. Given that you do expect tapirs to do so, the fact that FIT did indicate so is a reassuring outcome but it is not proof so much as consistency with expectations. FIT, as with any estimation procedure, provides confirmation on test data but inference on field data.

Line 245: Again, the FIT analysis infers long-range movement, it does not demonstrate it, in the same was as say radio tracking would. Footprints are no 100% unambiguously identifiable so you do not have unambiguous evidence of such movement, but rather inference based on an estimation procedure. I emphasize this point not to undervalue the approach but to be precise.

Line 250: This result indicates…

Line 261: The extent to which the dirt roads sample the tapir habitat is an important one. Can you say a little more about whether the 70 dirt roads would provide adequate coverage to sample tapir activity using FIT in this area? More generally, is FIT practical on other than dirt roads for tapir, e.g., on game trails?

Line 268: ‘trackers’ expertise’ or ‘tracker expertise’

Line 276: for the regression of tracks on tapir numbers the effects of sampling intensity and other factors might be incorporated into a more complex model. I agree it would require rigorous testing, though it may be useful as an index.

Lines 382 – 383: This citation is incomplete, the journal title is missing
In table 2, the entry ‘A+B” needs to be lined up with its data better.

Experimental design

A reader would need to consult the references provided by the authors on the method FIT employed in this paper to fully understand the method. Given the previous publications on this method, referenced and amply cited in this paper, I think this approach is appropriate in this paper.

Validity of the findings

Minor concerns regarding statements of conclusions are detailed in the minor revisions recommended to the authors in section 1.

Additional comments

The paper provides another example of the application of the FIT method, further demonstrating its utility for ecological and conservation science. The tapir is an interesting example since its footprint is rather unlike the other examples reported so far. The ecological role of the tapir and its challenging environment adds further value to this study.

Reviewer 2 ·

Basic reporting

In general the paper is well written and organized. Some editing is necessary for proper English. The following are some examples, but others are likely. Examples are keyed to line numbers:
18 - populations not population; ecosystem not ecosystems
42 - in South not in the South
46 - “lowland tapir populations are” not “the lowland tapir population is” unless you saying that there is only one population in Brazil
53 - “like” is not correct usage
54 - plant not plants
61 - individuals not individual
67 - is essential not are essential; “tapirs are” not “the tapir are”
99 - “to census” not “to censusing”; but actually, a census implies a complete enumeration of individuals
233 - data were not data was
268 - trackers’ not trackers
269 - species’ not species

Other items that need correcting/changes:
25 - you use “Footprint Identification Technique” here and in lines 3 and 82 (maybe elsewhere); the picture of the webpage as well as Table 2 uses “Technology” not “Technique”; seems surprising given that the software designers are listed as authors
79 - actually, tapir images from camera traps can sometimes be separated to individual on the basis of scars, scratches, and other marks; such marks may not necessarily be permanent but may be sufficient for separate sample periods
90-94 - scientific names of all additional species are needed
138 “after dusk and dawn” is vague

Table 2 - says there were 440 total footprints and 46 total trails; the results (line 215) says 547 footprints and 48 trails; also, the numbers in parentheses are not explained in the table heading; the column heading “Estimated number of tapir” should be “Estimated number of tapirs”; the data for line A+B are offset one line up

Experimental design

The description of the methods is clear and well described; there are no experiments per se. The results of the study depend on the validity of the technique used (FIT). From the literature cited, it appears that the method has been used for a variety of species. It also seems that the software designers were part of virtually all the studies. To be completely convincing, some independent tests of the technique would seem reasonable (i.e., can the same results be obtained by someone else); on a related note, it would be good to know if the software will be made available - either open access or for a license fee.

Validity of the findings

Assuming the technique is correct, then the results appear reasonable for this particular site. As mentioned above, some independent tests would be useful. On lines 183-187, the method description states that 15 landmark points are selected - if different people select the landmarks, are 1) the same landmarks always selected and 2) are the results the same. From the description of the site and the images provided, the road surfaces look fairly ideal for tracks. On the other hand, tapirs in most areas likely do not have the option to walk along roads and, instead, walk along trails within forests. Often those trails may be very different surfaces, frequently muddy and frequently of different “hardness”. Consequently, tapir tracks in muddy surfaces would look very different. To extend the validity of this study beyond the specific site, comparisons with results from different sites and different surfaces would be necessary.

Additional comments

It is stated that the FIT allows one to distinguish individuals, sex, and age but only number of individuals is reported. Were you able to distinguish sex and age and, if so, why not report those numbers (if you were not able to, it would be useful to know that as well). On a somewhat related point, on lines 280-282, you suggest that a goal is a long-term study on tapirs in the area (a very valuable goal, indeed). A long-term study would, presumably, cover individuals from young to adult ages, with, presumably, considerable changes in footprints - do you know if an individual’s footprint can be distinguished no matter the age? or do the footprint characteristics change with age? This would be important for long-term demographic studies.

Lines 259-260 - This is an important caveat: “With appropriate weather conditions and substrates, and a moderate sampling effort, footprints can be used effectively for censusing tapir populations.” It might be useful to expand a bit on this point - particularly with respect to substrates that tapirs typically encounter.

---

## Round 0.2 · Minor Revisions

You should include in your final version some comments about the independent tests of the FIT technique.

One of the reviewers said: "The results of the study depend on the validity of the technique used (FIT). From the literature cited, it appears that the method has been used for a variety of species. It also seems that the software designers were part of virtually all the studies. To be completely convincing, some independent tests of the technique would seem reasonable (i.e., can the same results be obtained by someone else); on a related note, it would be good to know if the software will be made available - either open access or for a license fee." In your rebuttal, you said, "The FIT software is being now used by independent research groups—e.g., for Eurasian river otter and cheetah monitoring. We work in collaboration with independent research groups, in this case, tapir research groups in Brazil. The FIT is provided by Wildtrack free of charge usually at the end of a training workshop to ensure that the end-user is able to use the technique to maximum effect."

However, you have not addressed this issue in the manuscript.

To have your paper accepted, you need to include more information in the manuscript about this specific issue pointed out by the reviewer.

---

## Round 0.3 · accepted · Accept

I believe you have addressed all comments by the reviewers. Congratulations again!